# The Diagnosis of Small Gastrointestinal Subepithelial Lesions by Endoscopic Ultrasound-Guided Fine Needle Aspiration and Biopsy

**DOI:** 10.3390/diagnostics12040810

**Published:** 2022-03-25

**Authors:** Masanari Sekine, Takeharu Asano, Hirosato Mashima

**Affiliations:** Department of Gastroenterology, Jichi Medical University Saitama Medical Center, 1-847 Amanuma-cho, Omiya-ku, Saitama City 330-8503, Japan; asanota@jichi.ac.jp (T.A.); hmashima@jichi.ac.jp (H.M.)

**Keywords:** subepithelial lesion, EUS-FNA, EUS-FNB, biopsy, gastrointestinal stromal tumor

## Abstract

Endoscopic ultrasonography (EUS) has been widely accepted in the diagnosis of all types of tumors, especially pancreatic tumors, lymph nodes, and subepithelial lesions (SELs). One reason is that the examination can provide a detailed observation, with tissue samples being immediately obtained by endoscopic ultrasound-guided fine needle aspiration (EUS-FNA). Many SELs are detected incidentally during endoscopic examinations without symptoms. Most SELs are mesenchymal tumors originating from the fourth layer, such as gastrointestinal stromal tumors (GISTs), leiomyomas, and schwannomas. GISTs are potentially malignant. Surgical treatment is recommended for localized GISTs of ≥20 mm. However, the indications for the diagnosis and follow-up of GISTs of <20 mm in size are controversial. There are several reports on the rapid progression or metastasis of small GISTs. Therefore, it is important to determine whether a SEL is a GIST or not. The main diagnostic method is EUS-FNA. Recently, endoscopic ultrasound-guided fine needle biopsy (EUS-FNB) using a new biopsy needle has been reported to obtain larger tissue samples. Additionally, various biopsy methods have been reported to have a high diagnostic rate for small GISTs. In local gastric SELs, regardless of the tumor size, EUS can be performed first; then, EUS-FNA/B or various biopsy methods can be used to obtain tissue samples for decision-making in relation to therapy and the follow-up period.

## 1. Introduction

Subepithelial lesions (SELs) were previously referred to as submucosal tumors (SMTs). However, many of these lesions do not arise from the submucosa and are not tumors [1,2,3,4]. Thus, the term “SEL” is more appropriate than SMT.

Many SELs are detected incidentally during endoscopic examinations in patients without symptoms. Endoscopic ultrasonography (EUS) or an ultrasonic miniprobe is used as the next examination. A previous study reported that the diagnostic accuracy of EUS imaging alone was as low as 43% in SELs originating in the third and fourth layer [1]. Nishida et al. reported that screening endoscopy may reveal small SELs in 0.15% of middle-aged adults [5].

Many guidelines recommend the histological evaluation of SELs of ≥20 mm in size. These are often evaluated by endoscopic ultrasound-guided fine needle aspiration and biopsy (EUS-FNA/B), or various other biopsy methods. EUS-FNA is the gold standard tissue sampling method for SELs. However, a previous meta-analysis reported that the pooled diagnostic rate of EUS-FNA for SELs was only 59.9% [6]. Therefore, the guidelines recommended follow-up for small SELs of <20 mm in size. The main reason is that the diagnostic rate of EUS-FNA is poorer for these small SELs [7,8]. The surveillance and management of small SELs is especially controversial because the natural history of small gastrointestinal stromal tumors (GIST) is unknown.

We reviewed the indications and usefulness of EUS-FNA/B for small gastric SELs.

## 2. Endoscopic Features

The macroscopic shape of SELs can be demonstrated using the Yamada classification [9,10]. In Y-I (elevation with a smooth baseline without a clear boundary) or Y-II (elevation with a boundary at the base but no notch), the tumor arises from the deep submucosal or muscularis propria layer (e.g., GISTs, leiomyoma, and schwannoma). In Y-III (elevation with a clearly notched base but no peduncle) or Y-IV (pedunculated elevation), the tumor arises from the deep mucosal or superficial submucosal layer (e.g., inflammatory and granular cell tumors). Most SELs show a normal mucosal color, except lipoma and granular cell tumors are yellowish in color, and varices are bluish in color [11,12]. Most SELs are covered by a normal mucosa with a smooth surface. Sometimes, SELs present with central dimpling, erosion, or ulceration.

Using a biopsy forceps, the endoscopist can evaluate the softness based on the cushion sign and mobility based on the rolling sign. The rolling sign indicates that the tumors originate from either the muscularis mucosa or deeper layers. Although not applicable to all SELs, the gastric region tends to be the predominant site of SELs. GISTs are commonly detected in the fundus and body, leiomyomas are commonly detected in the cardia and upper body, and schwannomas are commonly detected in the body [13,14]. Lipomas and ectopic pancreas are detected in the antrum [11,15].

## 3. EUS Imaging

EUS is a key examination for the differential diagnosis of SELs because it is a safe and useful modality that utilizes high-frequency ultrasound. Before we perform EUS-FNA, we need to diagnose SELs based on the findings of EUS images.

Kida et al., reviewed the characteristic findings of SELs [16]. With respect to the originating layer, neuroendocrine tumors originate from the second and third layers, lipomas, lymphangiomas, fibromas, and Brunner’s gland hyperplasia originate from the third layer, ectopic pancreas originates from the third and fourth layers, and GISTs, leiomyomas, and schwannomas originate from the fourth layer.

With respect to the echo level, GISTs, neuroendocrine tumors, leiomyomas, leiomyosarcomas, schwannomas, and ectopic pancreas are detected as a slightly hypoechoic to isoechoic tumor, while lipomas appear as hyperechoic tumors.

However, in the diagnosis of the EUS images, it is difficult to differentiate GIST from leiomyoma and schwannoma. In SELs, GIST is the most common mesenchymal tumor and has malignant potential. Surgical treatment should be indicated for GISTs. High-risk EUS features of GIST include irregular borders, heterogeneous echo pattern, anechoic spaces, and echogenic foci, but the tumor size cannot be used to predict the risk of malignancy [8]. We reported that there was no difference in the features of EUS images between lesions of <20 mm in size and lesions of ≥20 mm in size [17].

Among other methods, contrast-enhanced EUS (CE-EUS) has been reported to be useful. CE-EUS was established to assess the perfusion of SELs by high-resolution harmonic imaging using lower acoustic power. CE-EUS was able to differentiate between low-grade and high-grade malignant potential, including the microvasculature, parenchymal perfusion, and non-enhanced spots. Using these imaging features, CE-EUS showed increased sensitivity [18,19]. Sakamoto et al. reported that even small GISTs have malignant potential and can be detected by CE-EUS because irregular vessels observed by CE-EUS were more sensitive in the evaluation of malignant GIST than an irregular extraluminal border or necrotic appearance on standard EUS, especially in the case of small GISTs [20].

## 4. Types of SELs

### 4.1. Mesenchymal Tumors

Mesenchymal tumors of the GI tract are mainly classified into three types: GIST, leiomyoma, and schwannoma. Pathologically, most of these tumors are composed of spindle cells, or partially show smooth muscle or nerve sheath differentiation [21,22].

#### 4.1.1. GISTs

GISTs are mesenchymal tumors and have malignant potential. GISTs originate from the interstitial cells of Cajal. Most GISTs look like spindle cell tumors (70%), but they can be epithelioid (20%) or mixed type (10%). Immunohistochemistry is important for differentiation from other mesenchymal tumors. Seventy percent of GISTs stain positive for CD34, which was the first reported immunohistochemistry marker. C-kit (CD117) is a more sensitive and specific marker. GIST-1 (DOG1) is a novel marker that is expressed irrespective of the C-kit or platelet-derived growth factor receptor α (PDGFRA) mutation states [23]. More than 95% of GISTs stain positively for both C-kit and DOG1.

#### 4.1.2. Leiomyomas

Leiomyomas are benign tumors that originate from either the muscularis mucosae or muscularis propria. The EUS imaging features of leiomyomas are similar to those of GISTs. Recently, a predictive nomogram for differentiating GIST and leiomyoma was reported [24]. The nomogram includes the following parameters: age, sex, homogeneity and anechoic space (EUS features), location in the stomach, and ulceration. The nomogram maybe helpful for differentiating GISTs from leiomyoma at sizes of <20 mm, and 20–35 mm. Leiomyomas are diagnosed based on their histology (desmin-positive, α-SMA -positive, C-kit-negative, and CD34-negative).

#### 4.1.3. Schwannomas

Schwannomas have very low malignant potential. These tumors are composed of spindle cells originating from any nerve with a Schwann cell sheath (e.g., Auerbach’s plexus or Meissner’s plexus). On EUS imaging, schwannomas look similar to GISTs. Histology (S100-positive, C-kit-negative, and CD34-negative) is necessary for the diagnosis of schwannoma.

### 4.2. Neuroendocrine Neoplasm

Neuroendocrine neoplasm (NEN) is often seen in the gastrointestinal tract, for example, the stomach, duodenum, and rectum. NEN shows a hypoechoic pattern in EUS. NEN arises from a deep second or third layer and can be diagnosed by endoscopic biopsy. Rindi classification is used for gastric NEN [25]. Rindi classified gastric NEN into three types. Type I is caused by hypergastrinemia due to the gastric mucosa atrophy. Type II is associated with hypergastrinemia from Zollinger-Ellison syndrome. Type III arises sporadically and is independent of gastrin. Type I, II and III have low, moderate, and high risk of metastasis, respectively. Therefore, the diagnosis of NEN is as important as that of GIST.

### 4.3. Malignant Lymphoma

Primary gastrointestinal lymphoma is very rare. The majority of the primary gastrointestinal lymphomas are of B cell origin, for example, mucosa-associated lymphoid tissue (MALT) lymphoma in the stomach and mantle cell lymphoma in the jejunum and colon. The stomach is the most commonly involved site in the gastrointestinal tract. Gastric lymphoma constitutes 3–5% of all malignant tumors of the stomach [26]. *H. pylori* plays a role in the development of most MALT lymphoma. The endoscopic features include ulceration, diffuse infiltration, and polypoid lesion. Malignant lymphoma in stomach sometimes shows a SEL-like appearance but can be diagnosed by endoscopic biopsy. In EUS, the lymphoma is assessed usually hypoechoic lesion. EUS can detect the depth of lymphomatous invasion and the presence of perigastric lymph nodes [27].

### 4.4. Ectopic Pancreas

Ectopic pancreas is defined as pancreatic tissue without anatomical or vascular connection to the pancreas. Ectopic pancreas, also called heterotopic or aberrant pancreas, is not associated with any symptoms but sometimes causes gastrointestinal bleeding [28]. Using EUS, the depth of ectopic pancreas can be classified into two types: shallow type and deep type. Ectopic pancreatic tissue originates from the third layer in the shallow type and from the third and fourth layers in the deep type. In 1909, Heinrich classified ectopic pancreas into three types. Heinrich I contains all components, including acinar cells, a ductal structure, and islets of Langerhans; this type is generally large and exists in the middle and upper part of the stomach. Heinrich II contains an incomplete or lobular arrangement and lacks an endocrine component. Heinrich III contains ectopic tissue of proliferating ducts, and does not exhibit either acinar cells or endocrine components [29]. Characteristic EUS features of ectopic pancreas include an indistinct margin, heterogeneous echogenicity, and sometimes a ductal structure and cystic components.

Malignant transformation of ectopic pancreas has been reported, the associated features include male sex, middle age, location in the stomach, Heinrich I, and subepithelial-like macroscopic appearance. Malignant transformation of ectopic pancreas is rare, and it has an improved prognosis in comparison to pancreatic cancer, however, it should be considered to have malignant potential [30].

### 4.5. Lipomas

Lipomas are benign tumors composed of mature lipocytes. On endoscopy, lipomas usually present as solitary yellowish lesions, and are soft and positive for the cushion sign. EUS demonstrates that lipomas originate from the third layer and show a hyperechoic pattern. As a result, diagnosing lipoma with EUS tends to be rather easy, and therefore EUS-FNA is not considered to be necessary.

### 4.6. Varices

Occasionally, large gastric varices may be polypoid [31]. EUS imaging of gastric varices demonstrates characteristic anechoic serpiginous structures in the third hyperechoic layer. Flow within the varix can be demonstrated by a Doppler examination.

### 4.7. Glomus Tumors

Glomus tumors are mesenchymal tumors that originate from modified smooth muscle cells of the glomus body [16,32]. They are rarely found in the stomach. EUS demonstrates that glomus tumors arise from the fourth layer, or sometimes the third and fourth layers. The tumors are usually isoechoic and homogeneous, but are sometimes hypoechoic and heterogeneous. The diagnosis of glomus tumors is based on the histology (SMA-positive, vimentin-positive, laminin-positive, C-kit-negative, and CD34-negative) [33].

### 4.8. Metastatic Malignant Tumor

Gastric metastases have been reported from carcinomas such as breast, lung, kidney, ovary, and cutaneous melanoma. The common macroscopic feature of gastric metastasis is SEL-like, with diffuse infiltration of tumor cells to the submucosa and muscularis propria. The EUS images depend on the origin of the tumor and are not uniform. Metastatic tumors can usually be diagnosed by endoscopic biopsy, but EUS-FNA/FNB may be required.

## 5. Indications

EUS-FNA/FNB should be performed when the results of the tissue diagnosis may change the choice of treatment. The selection of surgical procedure and plan may be changed according to the findings of histology.

For the accurate diagnosis of SELs, it is important to obtain a sufficient amount of tissue to perform an immunohistochemical examination. The amount of tissue collected is affected by various factors, including the type of tumor and tumor size.

Patients with locally advanced GISTs and metastatic GISTs have a 5-year survival rate of 80% and 55%, respectively [34]. For the diagnosis of GISTs of ≥20 mm in size, it is recommended to obtain histological specimens by EUS-FNA or surgery. When Miettinen et al. categorized the malignant risk of GISTs, gastric GISTs of <20 mm in size were categorized into the low-risk group, [35] as all guidelines recommend EUS followed by periodic surveillance for small SELs (<20 mm) [36,37].

On the other hand, some medical specialists recommend that the optimal treatment for localized GIST is surgery, regardless of the tumor size. The Canadian guidelines recommend that GISTs, even those of <10 mm in size, be resected because of the risk of metastasis [38]. Additionally, some authors have reported that small GISTs without high-risk features showed rapid progression or metastasis [39,40].

The diagnosis of small SELs by EUS-FNA is difficult [41]; thus, the indication of EUS-FNA for small SELs (<20 mm) is controversial.

## 6. Clinical Course

Most small SELs do not increase in size. GISTs are potentially malignant, but the follow-up interval for small GISTs is controversial. Several studies have reported the follow-up period of small GISTs. Ye et al. reported that the majority of small (<20 mm) suspicious gastric GISTs showed no increase in size (98% 402/410), and that only 8 lesions (2%) increased in size during a median follow-up period of 28 months [42]. We also reported one difference: there was no significant increase in tumor size of small (<20 mm) GISTs during a median follow-up of 66 months, while tumors of ≥20 mm tended to become larger [17]. Most small GIST can be observed, and EUS-FNA and surgery may be performed when they increase to ≥20 mm. Metastasis or invasion of GISTs of <20 mm in diameter is considered very rare [43,44].

However, there were rare cases describing metastasis or the rapid progression of small GIST. Aso et al. encountered a patient with metastasis from a GIST of approximately 15 mm in diameter [40]. We also experienced one case in which a gastric GIST showed rapid progression, with the tumor size increasing from 18 to 84 mm over 2 years, and the development of liver metastasis [17].

It is important to clearly identify whether a lesion is benign, malignant, or possesses a malignant potential, regardless of size. Small benign SELs do not increase in size, while malignant SELs or SELs with a malignant potential require a close and careful follow-up.

## 7. Diagnostic Yield of EUS-FNA/B

EUS-FNA is the gold standard tissue sampling method for SELs. The diagnostic rate of EUS-FNA for SELs ranges from 62% to 93% [45,46,47,48]. The diagnostic rate depends on the tumor diameter (1–2 cm, 71%; 2–4 cm, 86%; and >4 cm, 100% [6]. In cases involving small SELs (<20 mm), EUS-FNA sometimes fails to obtain a histological specimen, mainly because the material is insufficient [49]. This leads to failure in making a diagnosis, despite time-consuming procedures. Varadarajulu et al., reported that the presence of an on-site cytopathologist is considered to be a key factor for diagnostic sensitivity in EUS-FNA [50]. The diagnostic yield of EUS-FNA with a ROSE is reported to be 100% for gastric SELs regardless of the tumor size [51].

Recently, it is reported that EUS-FNB using a new biopsy needle (franseen, reverse-bevel, or fork-tip needles) (Figure 1), allows for the acquisition of a greater amount of tissue. Many reports have indicated that EUS-FNB is more useful than EUS-FNA [52,53]. The details of studies on EUS-FNA/B are shown in Table 1.

Clinical trials have evaluated the diagnostic rate of EUS-FNA only, EUS-FNB only, and both EUS-FNA and EUS-FNB [58,59,60,61,62]. Most trials that evaluated the diagnostic rate of EUS-FNA only compared EUS-FNA with various biopsy methods. Most studies reported that FNB showed a superior diagnostic rate to FNA, because FNB achieved adequate sampling.

The diagnostic rate of EUS-FNA for lesions of any size, <20 mm, and ≥20 mm was 34.8–82%, 39.1–83.3%, and 33.3–91.6%, respectively. On the other hand, the diagnostic rate of EUS-FNB for lesions of any size, <20 mm, and ≥20 mm was 64.3–94.1%, 46.7–100%, and 77.7–100%, respectively. Regarding the number of needle punctures, three passes of EUS-FNB without ROSE showed good sensitivity (84%) [63].

Regarding the location of SELs, puncture of the gastric body is easy, while puncture of the fundus and antrum is difficult because the fundus is difficult to stably hold a scope [63], while the antrum wall is thicker than the body and fundus [64].

The diagnostic rates of each institution varied. In general, the diagnostic rate of tumor size ≥20 mm tended to be superior to that of <20 mm, and EUS-FNB was superior to EUS-FNA. In these studies, the diagnostic rate of EUS-FNA tended to be low because the influence of technical skill might play a strong role in making an accurate diagnosis, with differences between institutions. In EUS-FNB, there tended to be less difference in the diagnostic rate; thus, the influence of technical skill may be small. Fujita et al. reported that trainee endoscopists could obtain adequate samples using EUS-FNB [59].

Typically, the technical success rate for lesions of ≥20 mm in diameter was higher than that for lesions <20 mm. For SELs of <20 mm diameter, the diagnostic rate of EUS-FNB was higher than that of EUS-FNA. Both procedures were associated with low rates of adverse events.

The various reports showed the rate of adverse events of EUS-FNA/B was low. In adverse events of comparing between 22 gauge needle and 25 gauge needle, a meta-analysis and systematic review showed no significant difference [65,66]. The rate of adverse events showed no significant difference between the FNA needle and the FNB needle. And all cases of adverse events were mild, as three bleeding of EUS-FNB, two bleeding of EUS-FNA, and one aspiration pneumonia of EUS-FNA [67].

Recently, Forward-viewing, curved linear-array, and drill needles are expected to improve the diagnostic rate for small SELs [56,68]. EUS-FNA using Forward-viewing EUS is safe and feasible for small SELs, and a method with a cap device and suction is more useful.

## 8. Comparison with Various Biopsy Methods

The efficacy of conventional endoscopic forceps biopsy is limited because the forceps can only grasp the surface layer and therefore the tumor cannot be reached. However, biopsy within an ulcer is effective when ulceration is present [69]. The diagnostic yields of jumbo biopsy or bite-on-bite biopsy (multiple biopsy specimens obtained from the same site) are also poor, ranging from 17% to 59% [54,55,57].

Recently, Various biopsy methods using Endoscopic submucosal dissection (ESD) techniques have been reported, especially for small SELs (<20 mm). The details of studies on various types of biopsy are shown in Table 2. Minoda et al. showed that for small (<20 mm) lesions, mucosal incision-assisted biopsy (MIAB) showed a higher tissue sampling success rate in comparison to EUS-FNA/B (93.3% vs. 71.4%, respectively) [61]. In the MIAB method, first, normal saline or glycerol is injected into the submucosal layer to lift the mucosa covering the SEL. The target mucosal and submucosal tissues are cut using an endoscopic submucosal dissection knife with a high-frequency power supply. After exposing the lesion, 3–7 biopsy samples are obtained with a biopsy forceps. The diagnostic rate of MIAB for lesions of <20 mm and ≥20 mm was 80–93.3% and 91.7–92.3%, respectively [61,70,71].

Regarding other methods using ESD techniques, mucosal cutting biopsy (MCB) and submucosal tunneling biopsy (STB) have been reported. In MCB, the mucosa is incised 10–15 mm in length after saline is sometimes injected into the mucosa. Next, the tissue sample is collected using biopsy forceps from the exposed tumor. Finally, the wound is closed with endoclips. In STB, a 10 mm opening flap and a short tunnel is created to approach the lesion. Next, the tissue is collected by forceps. Finally, the entry is closed with endoclips. The diagnostic yield of MCB was 72–86% for lesions of <20 mm in diameter [72,73], and that of STB was 100% for lesions of any diameter [74]. Unroofing biopsy and deep biopsy were performed using a snare with electric current. The diagnostic rates of unroofing biopsy and deep biopsy for lesions of <20 mm were 80% and 93.8%, respectively [75,76]. All methods showed a high diagnostic rate and low rate of adverse events and these techniques do not require advanced endoscopy training [77].

Although EUS-FNA is the gold standard for obtaining tissue specimens for the histological analysis of suspicious gastric GISTs, various biopsy methods, stated above, can be used as alternative methods for obtaining tissue specimens with an intraluminal growth pattern [70].

## 9. Treatment

After the diagnosis of GIST, one therapeutic option is surgical resection for GIST without metastasis, another option for GIST with metastasis or recurrent GIST is to administer a tyrosine kinase inhibitor (TKI), such as imatinib [78,79]. The current guidelines recommend surgical resection for GISTs with symptoms, GISTs of ≥20 mm in size, or GISTs with high-risk features [5,80]. However, in a sub-analysis of a large epidemiological study, patients with GISTs of <20 mm who were observed did not show a significant difference in disease-specific mortality from those treated with surgical resection [81]. Surgical resection is recommended for small GISTs in the Japanese and Asian guidelines [78,82], and observation is recommended (excluding cases with high-risk features) based on empirical evidence in the NCCN guidelines [83]. The high-risk features of small GIST need to be reevaluated, and the surgical indications for small GIST should be reconsidered.

In previous studies, laparoscopic resection for GIST was reported to be safe, feasible, cost-effective, and superior to open resection [84,85,86]. Additionally, it is reported that several treatment methods using ESD techniques, such as submucosal tunneling endoscopic resection (STER), endoscopic full-thickness resection, laparoscopic endoscopic cooperative surgery (LECS), and non-exposed endoscopic wall-inversion surgery (NEWS), and a combined laparoscopic and endoscopic approach for neoplasia with a non-exposure technique (CLEAN-NET) have shown good clinical outcomes [87,88,89,90].

The feasibility and safety of laparoscopic resection has been established, but small size or intraluminal lesions are difficult to find laparoscopically. Endoscopic resection may be an alternative to surgical resection for removing SELs in some cases. However, endoscopic resection is associated with a high risk of perforation, because most SELs arise from the muscularis propria. STER is a modified procedure of perioral endoscopic myotomy (POEM) for esophageal achalasia, and the safety have been proven for SELs [91,92]. LECS combines laparoscopic resection and endoscopic resection, an advantage of LECS is that LECS can resect both the serosal and mucosal layers under direct visualization [93], but one disadvantage is that the connection between the stomach and abdominal cavity inevitably occur.

As a result, we must keep in mind the dissemination of malignant cells in the peritoneal cavity. To solve this problem, closed-LECS, NEWS and CLEAN-NET were applied. These procedures involve a non-exposure technique for the full-thickness resection of the gastric wall and also differ in how the resected specimens are retrieved [94]. In closed-LECS and NEWS, the resected specimen is retrieved from inside the stomach, while in CLEAN-NET, the specimen is retrieved from outside the stomach using laparoscopy. These procedures are associated with some difficulties. Closed-LECS and NEWS are limited to tumors smaller than 30 mm in size for per-oral retrieval. NEWS requires a long operation time. In CLEAN-NET, it is difficult to determine the resection line of lesions located at the cardia and posterior wall of the upper third of the stomach.

LECS for small-size tumors is difficult. On the other hand, ESD was reported to have a high complete resection rate (91.7%) and low complication rate (3%) for small SELs of <20 mm [89].

Hsiao et al. reported that a nomogram was useful for the selection of therapy [95]. The nomogram could predict the need for LECS. The parameters of the nomogram included age, sex, size, and site. Endoscopic resection requires high degree of skill for endoscopists, while LECS requires significant manpower and an operation room. The nomogram can be useful for discussions between endoscopists and the surgeon regarding treatment plan decisions.

## 10. Follow-Up

As asymptomatic SELs of <20 mm harbor a very low risk of progression and usually show a benign clinical course, these tumors can be followed up periodically at intervals of 6 months to 2 years [96,97].

In patients with local gastric GIST of ≥20 mm in size, surgery is the primary treatment. While conservative follow-up is suggested for local gastric GIST of <20 mm. However, there is no evidence on the optimal follow-up interval for small GISTs of <20 mm in size.

The recommendations for small GIST are remarkably different among the various guidelines [42,83,98,99]. The National Comprehensive Cancer Network, The Japan Gastroenterological Endoscopy Society, and French guidelines recommend follow up intervals of 6–12 months, 1–2 years, and 6 and 18 months, respectively, and then every 2 years. The European Society for Medical Oncology (ESMO) suggested initial EUS at 3 months after detection, and the follow-up interval extension in the case of no growth [100]. Gao et al. reported that the follow-up interval for lesions of ≥9.5 mm should be 6–12 months, while that for GISTs of <9.5 mm should be 2–3 years, because suspected GISTs of ≥9.5 mm may show significant progression [101].

In postoperative cases, the assessment of the risk of recurrence is important for management. The main targets are local recurrence, liver metastasis, and peritoneal dissemination. Contrast-enhanced computed tomography (CT) is useful, and recommended as a follow-up examination method in the Japanese clinical practice guidelines for GIST [78].

The following prognostic factors were established: tumor size, location, and mitotic count of tumor cells [83,102]. Several risk stratifications and nomograms have been reported, including the National Institutes of Health (NIH) consensus criteria, the Armed Forces Institute of pathology criteria, the modified NIH classification, and Gold’s nomogram [43,44,102,103]. In the stratification system, size was categorized as <2 cm, 2–5 cm, 5–10 cm, and >10 cm, and mitosis was categorized as <5/5 mm^2^, 5–10/5 mm^2^, and >10/5 mm^2^. Genotype of C-kit and PDGFRA, the presence of clinical symptoms, and histological necrosis have also been reported as possible prognostic factors [104,105]. For example, GISTs with deletion mutations of codons 557–558 of C-kit Exon 11 have been shown to have a poor prognosis, while most PDGFRA-mutated GISTs and Succinate dehydrogenase (SDH)-mutated GISTs show better prognoses. In the previous case reports of rapid progression and metastasis, the two small GISTs might have some molecular mutations, such as C-kit, PDGFRA, and B-Raf [17,40]. However, unfortunately, the two cases were not assessed in the molecular evaluation.

Additionally, the modified NIH classification includes tumor rupture as a prognostic factor [102,106]. Rupture was not universally defined in the clinical factors, because the criteria for tumor rupture were different. Recently, a universal definition of tumor rupture was proposed [107,108]. The composite definition of rupture includes tumor fracture or spillage, blood-stained ascites, gastrointestinal perforation at the tumor site, microscopic infiltration of an adjacent organ, piecemeal resection performed during either laparotomy or laparoscopy, or an incisional biopsy performed during either laparotomy or laparoscopy, in contrast, R1 surgery, intraluminal penetration of the tumor, transperitoneal needle biopsy without any adverse events, and peritoneal penetration of microscopic tumor cells in pathological examinations were not considered to be associated with tumor rupture. In fact, ESMO guideline indicate long adjuvant therapy for ruptured GISTs because they associated with very high risk of peritoneal relapse [36]. However, even with this definition, GISTs with low mitotic counts showed low recurrence rates, even in the presence of tumor rupture [109].

The clinical course of post-operative GIST is unknown. An observation interval of 6 months to 1 year is recommended for non high-risk cases, while an interval of 4–6 months is recommended for high-risk cases. Recurrence is infrequently observed after the first 10 years of follow-up; thus, follow-up observation after surgery should be considered for a period of 10 years.

Another important factor to be considered is imatinib sensitivity [104]. The guidelines do not recommend adjuvant therapy for PDGFRA D842V-mutated GISTs and GISTs without C-kit or PDGFRA mutations (Wild-type GIST) because of the relatively indolent nature as well as their lack of imatinib responsiveness [36,83,104]. Previous reports showed that adjuvant therapy for 3 years improves recurrence-free survival as well as overall survival among patients with high-risk GISTs in comparison to 1 year of adjuvant therapy [109,110]. The optimal duration of adjuvant therapy has not yet been established; thus, the rate of recurrence after the discontinuation of adjuvant therapy was very similar among patients who received adjuvant therapy for 1 year, 2 years, 3 years, and 5 years, suggesting that imatinib activities are cytostatic [111]. Besides, the activity of each drug is well correlated with gene mutations and alterations; thus, in the era of precision medicine, cancer genome profiling should be considered when treatments are used. A targeted gene panel analysis and whole-exome sequencing may provide potential targeted therapy for “wild-type GISTs” and GISTs that are refractory to conventional TKIs.

Recently, the selection of neoadjuvant TKI therapy using EUS-FNB sample was reported [112]. They evaluated DNA sequencing of C-kit and PDGFRA in pretreatment GIST tissue. The success rate of DNA sequencing by EUS-FNB was 95% (77/81). The preoperative therapy was expected to reduce tumor size before surgery.

## 11. Conclusions

EUS-FNA/B is a minimally invasive and effective method for the diagnosis of SELs. However, there were wide differences in the diagnostic rates between institutions. In small SELs, first, we perform EUS. Next, we consider EUS-FNA/FNB in the facilities that are good at performing EUS, and several modified biopsy methods in the facilities that are good at performing ESD. When performing EUS-FNA, the use of FNB needles should be considered. Even in small SELs <20 mm, there are reports of rapid progression and metastasis. Therefore, the follow-up period and therapy should be carefully considered depending on the malignant potential of each case.

## Figures and Tables

**Figure 1 diagnostics-12-00810-f001:**
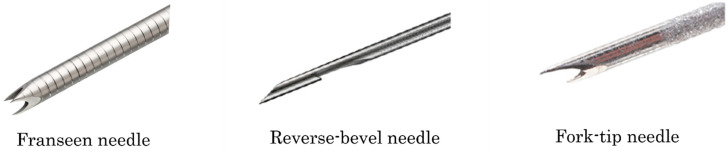
The shapes of the needles. (**Left**) Franseen needle. (**Center**): Reverse-bevel needle. (**Right**) Fork-tip needle.

**Table 1 diagnostics-12-00810-t001:** Details of studies on EUS-FNA and EUS-FNB.

Author	Year	FNA/FNB	Gauge of Needle	FNB Needle Used	Number of FNA	Number of <20 mm	Number of ≥20 mm	Diagnostic Rate of FNA	Diagnostic Rate of FNA <20 mm	Diagnostic Rate of FNA ≥20 mm	Number of FNB	Number of <20 mm	Number of ≥20 mm	Diagnostic Rate of FNB	Diagnostic Rate of FNB <20 mm	Diagnostic Rate of FNB ≥20 mm	Adverse Event (%)
Akahoshi et al. [42]	2007	FNA	22		51	21	30	0.82	0.71	0.90							0
Kobara et al. [54]	2017	FNA	19/22/25		23	17	6	0.35	0.35	0.33							0
Attila et al. [7]	2018	FNA	22		22	10	12	0.73	0.50	0.92							4.5
Adachi et al. [55]	2019	FNA	22		31	3	28	0.81	0.67	0.82							0
Osoegawa et al. [56]	2019	FNA	20/22/25		24	13	11	0.71	0.54	0.91							0
Park et al. [57]	2019	FNB	20/22	Reverse-bevel							28	15	13	0.64	0.47	0.85	0
Inoue et al. [38]	2019	FNB	19/20/22/25	Reverse-bevel, Franseen							57	30	27	0.82	0.67	0.97	3.5
Iwai et al. [51]	2018	FNA/FNB	19/22	Reverse-bevel	12	3	9	0.74	0.83	0.71	11	3	8	0.91	0.83	0.94	0
Fujita et al. [52]	2018	FNA/FNB	22	Franseen	44	15	29	0.75	0.60	0.83	17	5	12	0.94	1	0.92	0
Trindade et al. [53]	2019	FNA/FNB	19/22/25	Fork-tip	46	23	23	0.37	0.39	0.35	101	39	62	0.89	0.82	0.94	0
Minoda et al. [58]	2020	FNA/FNB	19/20/22/25	Reverse-bevel, Franseen	69	38	31	0.74	0.68	0.80	37	18	19	0.89	0.78	1	0
Sekine et al. [59]	2021	FNA/FNB	19/20/22/25	Reverse-bevel, Franseen	31	11	20	0.74	0.73	0.75	31	13	18	0.87	1	0.78	3.2 (FNB)

**Table 2 diagnostics-12-00810-t002:** Details of studies on various biopsy methods.

Author	Year	Procedure	Number of <20 mm	Number of ≥20 mm	Diagnostic Rate of <20 mm	Diagnostic Rate of ≥20 mm	Adverse Event (%)
Ihara et al. [68]	2013	MIAB	15	12	0.80	0.92	0
Osoegawa et al. [56]	2019	MIAB	11	12	0.91	0.92	0
Minoda et al. [58]	2020	MIAB	45	26	0.93	0.92	0
Adachi et al. [55]	2019	MCB	7	9	0.86	0.89	0
Nakano et al. [69]	2019	MCB	18	27	0.72	0.81	4.4
Kobara et al. [54]	2017	STB	29	14	1	1	0
Park et al. [57]	2019	unroofing biopsy	15	13	0.80	0.77	0
Abad-Belando et al. [70]	2018	deep biopsy	16	16	0.94	1	9.4

MIAB: mucosal incision-assisted biopsy; MCB: mucosal cutting biopsy; STB: submucosal tunneling biopsy.

## Data Availability

Not applicable.

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
