# Peer review of "The Diagnosis of Small Gastrointestinal Subepithelial Lesions by Endoscopic Ultrasound-Guided Fine Needle Aspiration and Biopsy"

_diagnostics, 2022, doi:10.3390/diagnostics12040810_

Round 1

Reviewer 1 Report

This review describes utilization of EUS-guided FNA and FNB on small gastrointestinal  subepithelial lesions (SELs). It is well written. I have a few minor points the authors might consider incorporating in the final form.

  1. I am surprised that the authors did not list some other common entities of SELs in the section of “Types of SELs.” These may include neuroendocrine tumors, lymphomas, and metastatic malignancies. Please add these entities and at least expand on discussion of neuroendocrine tumors.
  2. page 2 line 70, please change “Brunneriomas” to “Brunner’s gland hyperplasia.”
  3. Is there any follow-up molecular study on cases listed in reference 17 and 37 showing rapid progressing or metastasis?
  4. Cytohistological diagnosis of SELs is relatively straightforward. When comparing FNA with FNB, and comparing different FNB needles, have anybody looked into yield for molecular diagnosis rather than yield for histology diagnosis? This is more important for malignant neoplasms in which molecular studies may provide specific guidance in subsequent treatment. In addition, FNA may provide advantage in securing material for FLOW analysis when lymphoma is a top differential diagnosis.
  5. Please summarize or clarify “adverse events” associated with FNB. Different reports may count or define “adverse events” differently. In Table 1 are those adverse events more associated with larger needles? Is there any conclusion comparing 22g needle with 25g needle in terms of diagnostic yield and adverse events?  
  6. Please spell out “OS” and “ESD” the first time they are introduced.

Author Response

This review describes utilization of EUS-guided FNA and FNB on small gastrointestinal subepithelial lesions (SELs). It is well written. I have a few minor points the authors might consider incorporating in the final form.

  1. I am surprised that the authors did not list some other common entities of SELs in the section of “Types of SELs.” These may include neuroendocrine tumors, lymphomas, and metastatic malignancies. Please add these entities and at least expand on discussion of neuroendocrine tumors.

Answer: Thank you very much for pointing out missed important entities. We added these entities at P10 L13 – P12 L1 and P14 L14-P15 L2. NEN can usually be diagnosed by endoscopic biopsy, and we added the description in the section of “3. Types of SELs”

  1. page 2 line 70, please change “Brunneriomas” to “Brunner’s gland hyperplasia.”

Answer: We changed “Brunneriomas” to “Brunner’s gland hyperplasia.”

  1. Is there any follow-up molecular study on cases listed in references 17 and 37 showing rapid progressing or metastasis?

Answer: Unfortunately, the two cases were not assessed in the molecular evaluation. We added the details of the two cases at P28 L5-8.

  1. Cytohistological diagnosis of SELs is relatively straightforward. When comparing FNA with FNB, and comparing different FNB needles, have anybody looked into yield for molecular diagnosis rather than yield for histology diagnosis? This is more important for malignant neoplasms in which molecular studies may provide specific guidance in subsequent treatment. In addition, FNA may provide advantage in securing material for FLOW analysis when lymphoma is a top differential diagnosis.

Answer: Thank you very much for pointing out an important issue. We cannot find the study comparing FNA with FNB for molecular diagnosis. Recently, DNA sequencing of C-kit and PDGFRA using EUS-FNB sample has been reported. The success rate of DNA sequencing was as high as 95% and it may help the selection of neoadjuvant therapy. We added these descriptions at P30 L11-15.

  1. Please summarize or clarify “adverse events” associated with FNB. Different reports may count or define “adverse events” differently. In Table 1 are those adverse events more associated with larger needles? Is there any conclusion comparing 22g needle with 25g needle in terms of diagnostic yield and adverse events?  

Answer: We added the detail and the summarization of adverse events at P20 L5-11.

  1. Please spell out “OS” and “ESD” the first time they are introduced.

Answer: We spelled out the terms.

Reviewer 2 Report

Short and concise, good work.

Author Response

We are grateful for the reviewers' thoughtful comments and suggestions.

.

Reviewer2

Short and concise, good work.

Answer: Thank you very much for reviewing our manuscript.